# Structural resolution of inorganic nanotubes with complex stoichiometry

Geoffrey Monet[1], Mohamed S. Amara[1], Stéphan Rouzière[1], Erwan Paineau [1], Ziwei Chai[2], Joshua D. Elliott[3,5], Emiliano Poli[3,6], Li-Min Liu[2,4], Gilberto Teobaldi[2,3] & Pascale Launois [1]

Determination of the atomic structure of inorganic single-walled nanotubes with complex stoichiometry remains elusive due to the too many atomic coordinates to be fitted with respect to X-ray diffractograms inherently exhibiting rather broad features. Here we introduce a methodology to reduce the number of fitted variables and enable resolution of the atomic structure for inorganic nanotubes with complex stoichiometry. We apply it to recently synthesized methylated aluminosilicate and aluminogermanate imogolite nanotubes of nominal composition $(OH)_3Al_2O_3Si(Ge)CH_3$. Fitting of X-ray scattering diagrams, supported by Density Functional Theory simulations, reveals an unexpected rolling mode for these systems. The transferability of the approach opens up for improved understanding of structure–property relationships of inorganic nanotubes to the benefit of fundamental and applicative research in these systems.

[1] Laboratoire de Physique des Solides, UMR CNRS 8502, Université Paris Sud, Université Paris Saclay, 91405 Orsay Cedex, France. [2] Beijing Computational Science Research Centre, 100193 Beijing, China. [3] Stephenson Institute for Renewable Energy and Department of Chemistry, The University of Liverpool, Liverpool L69 3BX, UK. [4] School of Physics, Beihang University, 100191 Beijing, China. [5] Present address: Dipartimento di Fisica e Astronomia "Galileo Galilei", Università degli Studi di Padova, I-35131 Padova, Italy. [6] Present address: The Abdus Salam International Centre for Theoretical Physics, 34151 Trieste, Italy. Correspondence and requests for materials should be addressed to G.T. (email: G.Teobaldi@liverpool.ac.uk) or to P.L. (email: pascale.launois@u-psud.fr)

Single-walled nanotubes (SWNTs) constitute an appealing class of materials in which new synthesis strategies recently emerged[1–3]. Thanks to their one-dimensional properties and their large surface area, SWNTs are promising nano-bricks for applications in different fields, including nanoelectronics, nanofluidics, nanocatalysis, and selective molecular sieving[3–6]. Both organic and inorganic SWNTs are intensively studied for their complementary chemical and physical properties. With the discovery of single-walled carbon nanotubes (SWCNT), research in organic SWNTs grown exponentially[7,8] providing alternatives to supramolecular self-assembly and/or polymerization of nano-scopic organic systems[9–11]. Developments in inorganic SWNTs have been slower than for SWCNTs, with substantially fewer systems available until very recently, namely BN nanotubes[12], imogolite nanotubes (INTs)[13], $MoS_2$, $MoO_3$, and $SbPS_{4-x}Se_x$ nanotubes[14–16]. However, synthesis strategies have now emerged extending the family of inorganic SWNTs to sulfide, hydroxide, phosphate, and polyoxometalate nanotubes[1,3]. The generality of the recently proposed approach opens the way for future synthesis of a wide variety of inorganic nanotubes.

Knowledge of nanotubes' atomic structure is crucial for comprehension of their properties. The structure of SWCNTs is currently determined with a high level of accuracy, based in particular on electron diffraction experiments coupled to the theory of diffraction from helices, as well as on a recent powder X-ray scattering study[17–19]. Structural resolution of more complex organic nanotubes is usually based on wide-angle X-ray scattering (WAXS) by highly oriented nanotubes fibers[18] or, alternatively, oriented nanotubes in suspensions[2]. In ref. [20], oriented inorganic nanotubes suspensions could be obtained but with extremely low concentration. The lack of oriented samples of inorganic nanotubes, suitable for WAXS, is a major obstacle. Apart from BN nanotubes, whose structure in principle derives from that of carbon nanotubes by substituting alternatively C atoms by B and N ones, as well as the very special case of $MoS_2$ nanotubes that assemble in a crystalline three-dimensional system, there is no unambiguous and detailed determination of the atomic structure of inorganic SWNTs.

In this article we focus on accurate determination of the atomic structure of inorganic SWNTs from powder WAXS measurements. Powder X-ray scattering method has the advantage to be a statistical method, as compared to local analysis methods such as electron diffraction, and it does not require special sample preparation. The inorganic SWNTs considered here are the newly synthesized methylated metal-oxide INTs of nominal composition $(OH)_3Al_2O_3Si(Ge)CH_3$ (refs. [21,22]). We chose them as representative examples of inorganic nanotubes of complex stoichiometry as well as for their intrinsic properties.

Metal-oxide INTs with composition $(OH)_3Al_2O_3Si(Ge)OH$ were discovered in soils[23] and they can be easily synthesized using soft chemistry[24,25]. Their nanometric diameter is tunable at the Angstrom level, depending on the nature of synthesis precursors[22,26,27]. Moreover, various surface functionalizations have been achieved, either by modification of their inner surface with organic moieties[5,21,22] or by grafting organic compounds on the outer part of the nanotube[28,29]. The chemical versatility of INT paves the way towards possible applications in various fields, such as stimuli-responsive materials[20], molecular storage[21,30], molecular recognition and separation[5,31], water filtration and decontamination[22,32] as well as catalysis[33] and photocatalysis[34]. Furthermore, INTs are invoked in a geological context when evaluating carbon storage, metallic cations or radionuclides storage in soils[35–37], as well as markers in the evolution of the Martian climate[38].

Quantitative interpretation of imogolite WAXS diagrams has not been achieved yet, despite intensive research and while atomic positions for $(OH)_3Al_2O_3Si(Ge)OH$ INTs are provided in numerical simulations articles[39,40]. Neither careful analysis of synchrotron Pair Distribution Function[41] nor the comparison between experimental WAXS diagrams and calculated ones obtained by Tight-binding Density Functional Theory (TB-DFT) minimization[42] could lead to conclusive assignment of the atomic structure of the first inorganic SWNTs to be discovered, as early as 1962, namely aluminosilicate $(OH)_3Al_2O_3Si(Ge)OH$ INTs. It should be underlined that WAXS diagrams for nanotubes are not formed of Bragg peaks as in crystals. They consist of a limited number of broad modulations due to the nanometric lateral extent of the nanotubes. One cannot take advantage of three-dimensional crystalline order, as was done recently by Oda and co-workers[43] to solve the molecular structure of self-assembled organic nanoribbons.

A different approach is proposed here. It is based on the reduction of the number of independent atomic positions to be fitted to WAXS diagrams, thanks to the use of helical symmetries[44] and to a semi-empirical energy minimization. The strategy enables us to determine the atomic structure of both $(OH)_3Al_2O_3SiCH_3$ and $(OH)_3Al_2O_3GeCH_3$ nanotubes. The obtained structures are confirmed by DFT calculations.

## Results

**The structure of INTs.** The wall of aluminosilicate INTs and of their aluminogermanate analog consist of an octahedral gibbsite-like layer $(Al(OH)_3)$ with isolated $(Si(Ge)O_3)OH$ tetrahedron units connected via covalent bonding between three mutual oxygen atoms[13]. It can be first described with a three-dimensional $(OH)_3Al_2O_3Si(Ge)OH$ elementary unit arranged in a hexagonal lattice (Fig. 1). The graphene sheet, used to describe SWCNTs, exhibits the same hexagonal arrangement of C atoms. Following the convention adopted for SWCNTs[45], the structure

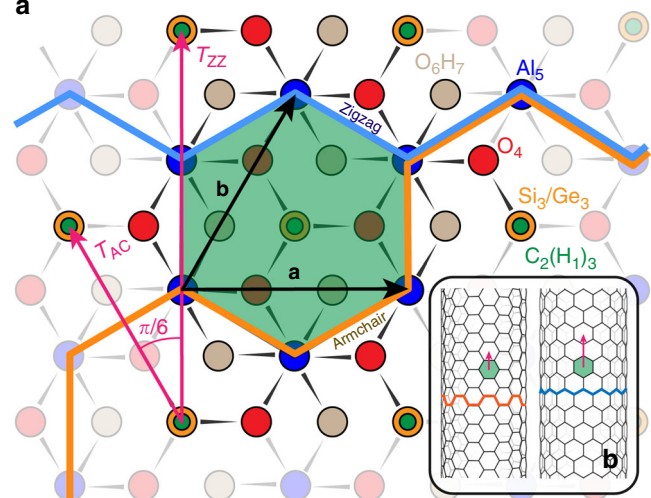

**Fig. 1** Imogolite unit cell. **a** Hexagonal unit cell of the imogolite nanotube or of its methylated analog. The color of each circle corresponds to a scattering center: green for inner OH ($CH_3$), gray for outer OH, blue for Al, red for O, and orange for Si/Ge. (**a**, **b**) is a lattice basis. The index refers to the radial atomic labeling introduced by Alvarez-Ramírez[47] and adapted by Poli et al.[34]. It corresponds to the sequence of atoms encountered on passing from the inner to the outer surface of the tube. The inset (**b**) displays an armchair and a zig-zag nanotube. Terms 'armchair' and 'zig-zag' are used in analogy with SWCNTs and illustrated by the orange and blue 'armchair' and 'zig-zag' lines drawn in Fig. 1 and in the inset. The periods of armchair and zigazg nanotubes, noted $T_{AC}$ and $T_{ZZ}$, are drawn with purple arrows

of an INT can be labeled by two integers $(N,M)$, the components of the so-called 'chiral vector' in the hexagonal basis (Fig. 2)[42]. The nanotube is obtained by cutting a ribbon perpendicularly to the chiral vector and eventually rolling it up. The strain energy of INTs presenting a well-defined minimum in diameter and in chirality[39,40,42,46–48], a macroscopic sample is expected to consist in nanotubes with the same $(N,M)$ indices. Accordingly, sharp diameter distributions are reported in the literature[20,22,26]. Current investigations of the structure of imogolite $(OH)_3Al_2O_3Si(Ge)OH$ nanotubes in the literature point towards a $(N,0)$ configuration, called 'zig-zag' (ZZ) by analogy with SWCNTs

(see Fig. 1), with a measured period $T_{ZZ} \approx 8.5$ Å along the tube axis[13,49–51]. But the experimental determination of the value of the index $N$ could not be achieved, as discussed in refs. [13,41,42].

In the methylated INTs (m-INTs) discovered a few years ago[21], inner hydroxyl groups are substituted by methyl groups, leading to a nominal composition $(OH)_3Al_2O_3Si(Ge)CH_3$. They are considered as ZZ nanotubes[21,22,30,34,52], like their hydroxylated analogs. Narrow diameter distributions are reported[22]. Available DFT and TB-DFT results suggest the hydrogen bonding network between inner hydroxyl groups in the pristine hydroxylated INTs to be key for the energetic favorability of the ZZ structure over an armchair (AC) one[39,40,53]. These results prompt careful investigation of the effects of methyl-substitution of the inner hydroxyl groups on the ZZ vs. AC energy competition, which we present here.

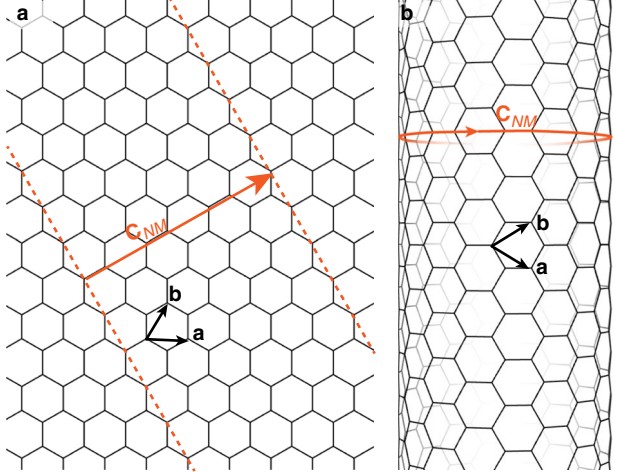

**Fig. 2** Rolling up and chirality of a honeycomb sheet. **a** A hexagonal sheet. **b** An example of a nanotube obtained by cutting a ribbon perpendicularly to the chiral vector and by rolling it up. The $(N,M)$ indices define the 'chiral vector' $\mathbf{C}_{NM} = N\mathbf{a} + M\mathbf{b}$ which joins two equivalent sites, $(\mathbf{a}, \mathbf{b})$ being a lattice basis. The norm of the chiral vector is equal to the nanotube perimeter and its orientation with respect to the basis vector ($\mathbf{a}$) defines the nanotube chiral angle

**WAXS experiments on m-INTs.** Using X-ray scattering experiments at relatively small wave vectors ($Q < 1$ Å$^{-1}$), where INTs can be approximated as homogeneous cylinders, Amara and co-workers[22] demonstrated that the inner and outer diameters of $(OH)_3Al_2O_3Si_xGe_{1-x}CH_3$ nanotubes decrease as $x$ increases from 0 to 1. However, no information about the atomic structure of the nanotubes could be obtained at such small wave vectors. In the present study and for the first time, we have performed powder WAXS measurements over a much wider Q-range, up to 8 Å$^{-1}$, for the two end-members $(OH)_3Al_2O_3SiCH_3$ and $(OH)_3Al_2O_3GeCH_3$ nanotubes, denoted SiCH$_3$ INT and GeCH$_3$ INT (Fig. 3). The recorded diagrams are made of rather broad modulations, which reflect the finite radial dimension of the nanotubes, together with more well-defined asymmetric peaks around 2.5–2.6, 5.1–5.2, and 7.6–7.7 Å$^{-1}$, as highlighted by arrows in Fig. 3a.

**Period values along m-INTs' axes.** The asymmetric peaks can be used to determine the period of the m-INTs along their long axis. Diffraction by any nanotube with a period $T$ along its long axis $z$ gives diffuse scattering intensity located in planes at $Q_z = l\frac{2\pi}{T}$,

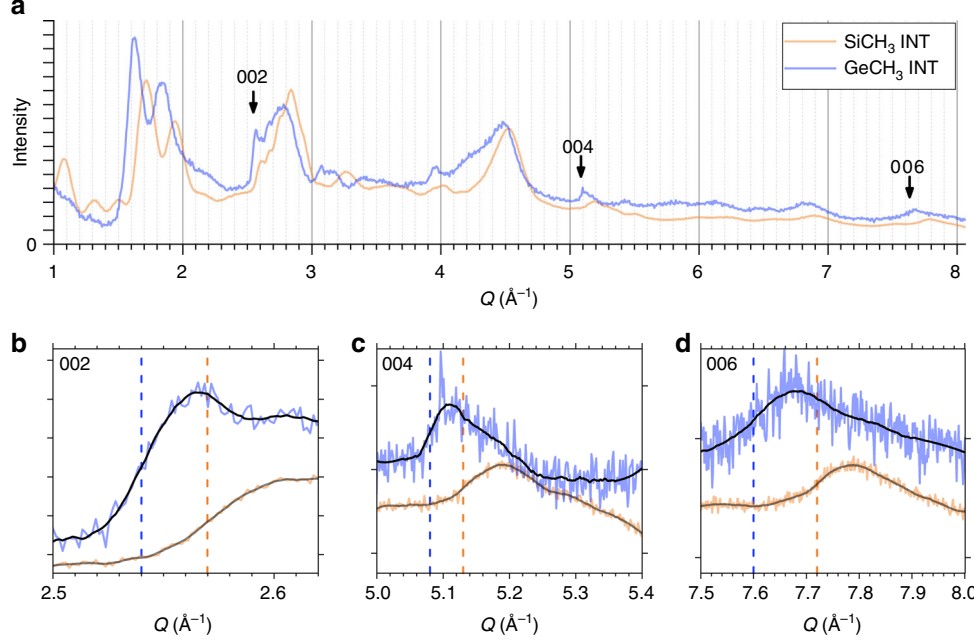

**Fig. 3** The period of methylated imogolite nanotube from powder WAXS diagrams. **a** WAXS of methylated Si(Ge)CH$_3$ INT powders. Insets **b**–**d** highlight 00$l$ asymmetrical peaks. Black curves are the result of the Savitzky-Golay filter, the number of points in the smoothing window is 15 (resp. 20) for 002, 40 (resp. 50) for 004, 60 (resp. 120) for 006 for SiCH$_3$ (resp. GeCH$_3$) INT. Ten points correspond to 0.01 Å$^{-1}$

**Table 1 Positions of inflexion points of 00$l$ sawtooth peaks from powder WAXS diagrams of Si(Ge)CH$_3$ INTs**

|  | Si-CH$_3$ | Ge-CH$_3$ |
|---|---|---|
| $Q_{002}$ (Å$^{-1}$) | 2.57 (3) | 2.54 (3) |
| $Q_{004}$ (Å$^{-1}$) | 5.13 (2) | 5.08 (7) |
| $Q_{006}$ (Å$^{-1}$) | 7.72 (10) | 7.60 (21) |
| Period $T$ (Å) | 4.89 (2) | 4.95 (3) |

The positions of inflexion points and the corresponding values of the period have been gathered and calculated from the Fig. 3. Uncertainty on last digits is given in parentheses

where $l$ is an integer[18]. When the scattered intensity is non-zero at wave-vector $\mathbf{Q} = \left(0, 0, Q_z = l\frac{2\pi}{T}\right)$, angular powder average leads to abrupt sawtooth peaks at $Q = l\frac{2\pi}{T}$ (ref. [54]). For nanotubes of finite length, these peaks are smoothed. Specifically, in the case of methylated nanotubes, which have typical lengths of the order of 100 Å (ref. [22]), these smoothing effect cannot be ignored. In this scope, the period value is not given by the position of the peak maximum but by the inflexion point of its rising edge[55]. To obtain inflexion point positions from the rather noisy scattering diagram, we applied a Savitzky-Golay filter with an adjustable window width as shown in Fig. 3b–d. Table 1 lists the corresponding Q-values. Assuming that these Q-values correspond to $l = 1$, $l = 2$, and $l = 3$ planes, one obtains $T = 2.45$ Å for SiCH$_3$-INTs and $T = 2.48$ Å for GeCH$_3$ INTs. However, such a small period is incompatible with the structure of the primary gibbsite sheet: the Al–Al distance in gibbsite is $d = 2.95$ Å (ref. [56]), so that the smallest period should be $d\sqrt{3}$, that is around 5 Å (see Fig. 1). The observed periodicity peaks have thus been indexed with $l = 2$, 4, and 6. The period $T$ is then found to be equal to 4.89 Å for SiCH$_3$ INTs and 4.95 Å for GeCH$_3$ INTs, respectively (Table 1). As noted above, previous studies[13,49–51] reported period values around 8.5 Å for aluminosilicate and aluminogermate (OH)$_3$Al$_2$O$_3$GeOH nanotubes. The determination of period values around 4.9 Å for methylated INTs gives a compelling evidence about chirality modification. The ratio between the periods of normal and methylated INTs is close to $\sqrt{3}$, which corresponds to the ratio of the periods between ZZ and AC structures (see Fig. 1). One will thus consider in the following methylated nanotubes (OH)$_3$Al$_2$O$_3$Si(Ge)CH$_3$ in AC configuration ($N,N$), in contrast with normal (OH)$_3$Al$_2$O$_3$Si(Ge)OH which present a ZZ configuration ($N$,0). The systematic extinction of 00$l$ reflections for odd $l$ values is then easily understood, since the period of an AC structure projected onto its long axis is equal to $T/2$ (see Fig. 1). It should be mentioned here that the assignment of a scattering maximum around 1.15 Å$^{-1}$ as corresponding to a $l = 1$ peak in ref. [57] is erroneous. No conclusion can be drawn about the nanotube chirality in ref. [57] because X-ray scattering diagrams were restricted to wave vectors smaller than 1.4 Å$^{-1}$, while the first periodicity peak is located around 2.5 Å$^{-1}$.

**Structure refinement from WAXS diagrams**. Thorough analysis of WAXS diagrams was undertaken to determine atomic positions. Knowing the period $T$, one should in principle refine atomic positions in a nanotube corona of height $T$, which contains hundreds of atoms, making WAXS fitting an underdetermined problem. For a ($N,N$) nanotube, fitted parameters can be reduced to the value of $N$, the positions of the atoms of the (OH)$_3$Al$_2$O$_3$Si(Ge)CH$_3$ elementary unit in Fig. 1 and to unit cell's parameters (modulus of unit cell vectors and angle between them), allowing unit cell distortions from the perfect hexagonal cell. Indeed, helical symmetries[44] allow one to generate a whole nanotube structure with inner radius $R_i$, outer radius $R_e$, and period $T$ from any planar unit cell. It should be noted here that

since X-rays are rather insensitive to H atoms, we considered virtual atoms at the electronic center of charge of OH and CH$_3$ groups, with X-ray form factors equal to the weighted sum of those of their constituents. To obtain structures of physico-chemical significance, we developed an algorithm allowing us to generate a full tubular atomic structure while minimizing an energy term $E_{geo}$. The subscript 'geo' stands for 'geometrical' as minimization is made over bond lengths and angles between them. The energy is calculated via a quadratic expansion over bond lengths and angles with relevant harmonic constants and reference bond lengths and angles values taken from the literature (Supplementary Notes 1 and 2). The total energy $E_{geo}$ is minimized with the Sequential Least SQuares Programming (SLSQP) algorithm with optional user-defined constraints like the inner and outer radii ($R_i$, $R_e$) as well as the value of the period $T$ determined on the WAXS diagram (Fig. 3b–d). Within this approach, one is left with only three parameters: $N$, $R_i$, and $R_e$. One may underline here the physico-chemical relevancy of such an approach. Values of inner and outer diameters, which reflect the role of inner and outer environments, typically aqueous environment during the synthesis, can be fixed if necessary (Supplementary Figure 5).

For a given set of parameters $N$, $R_i$, and $R_e$, an atomic file is generated over a relevant nanotube length $L$. It appears that $L = 100$ Å gives calculated WAXS diagrams in agreement with the experimental shape of the (00$l$) peaks (Supplementary Note 5). The powder WAXS diagram is calculated using Debye formula (Equation 1)[58]:

$$I(Q) \propto \sum_{i,j} f_i(Q)f_j(Q)\frac{\sin QR_{ij}}{QR_{ij}} \qquad (1)$$

where the sum runs over all pairs of atoms in the nanotube, $R_{ij}$ being the distance between atoms $i$ and $j$, $f_i(Q)$ and $f_j(Q)$ being the associated to scattering factors. Intensity calculations have been speed up using highly parallel calculation on GPUs[59]. On a regular laptop GPU (Nvidia GTX 860m), it takes about 1 h to fulfill a fitting procedure, i.e., to compute 2200 WAXS diagrams of 2048 Q-points from a structure comprising between 2840 and 5240 atoms depending on the value of $N$. A set of data calculated from powder X-ray scattering is created for a wide range of $R_i$, $R_e$, and $N$ values. Then, an algorithm extracts the one that matches the best the WAXS experimental data, between 1 and 8 Å$^{-1}$. This procedure is based on the identification and least-square fitting of eight well-defined maxima and of a minimum of the experimental diagram. The fitting procedure is summarized in Fig. 4 and it is detailed in Supplementary Note 3.

The best fit for SiCH$_3$ imogolite nanotube is obtained with $N = 9$, $R_i = 8.8$ Å, and $R_e = 13.6$ Å. The comparison between the calculated WAXS diagram and experiment is shown in Fig. 5a (Supplementary Note 3). GeCH$_3$ imogolite is slightly wider with (11,11) AC indices, $R_i = 11.6$ Å and $R_e = 16.2$ Å. Notice that wall thicknesses are in good agreement with the thickness of a fictitious imogolite planar structure in which a gibbsite layer is coated with Si/Ge tetrahedra (Supplementary Note 4). For both SiCH$_3$ and GeCH$_3$ nanotubes, CIF files are deposited at the Cambridge Crystallographic Data Centre; CCDC accession codes: 1838953, 1838955. No energy minimization having been performed for H atoms, their positions are arbitrarily chosen as follows. Hydrogen atoms of OH groups are radially lined up with O–Al bonds, and hydrogen atoms from methyl groups reproduce tetrahedra of a regular hydrocarbon.

The comparison between calculated and experimental WAXS diagrams appears satisfactory, especially since our simple proposed approach does not account for the possible existence of defects[60] and thermal disorder[61]. One may also notice that a

narrow distribution in diameter or chiral angle around the ones corresponding to the fitted (9,9) and (11,11) structures of SiCH₃ and GeCH₃ INTs cannot be ruled out (Supplementary Note 6).

Below 1 Å⁻¹, scattering patterns are more sensitive to the presence of water around nanotubes, as detailed by Amara et al.[22]. Subtle porosity effects can also be invoked in such Q-range. However, the signal in the 0.5–1 Å⁻¹ range, shown in Fig. 5b, allowed us to discriminate between configurations that appeared suitable considering only large-Q data (Supplementary Figure 4).

**DFT optimization of m-INTs' structures**. DFT optimization of the nanotube fitted geometry for the WAXS-derived period was

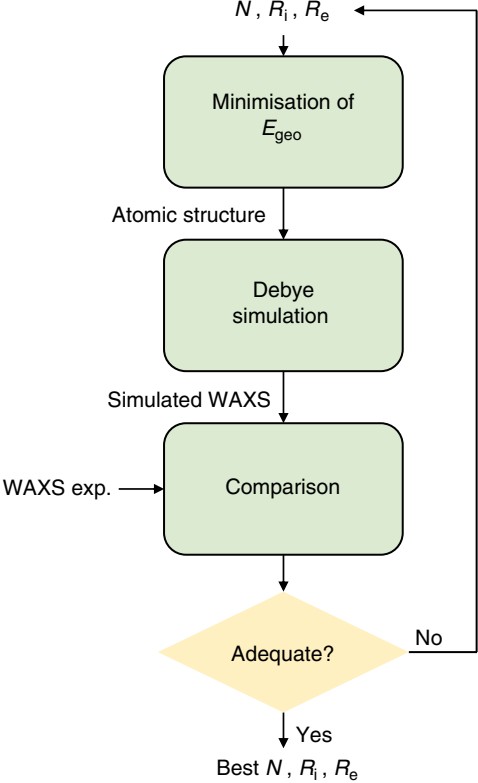

$N$, $R_i$, $R_e$

Minimisation of $E_{geo}$

Atomic structure

Debye simulation

Simulated WAXS

WAXS exp. ⟶ Comparison

Adequate?  No

Yes

Best $N$, $R_i$, $R_e$

**Fig. 4** Flowchart illustrating WAXS fitting method

performed both using the Perdew-Burke-Ernzerhof (PBE) functional and including dispersion interactions (PBE-D3). There is no significant difference between calculated WAXS diagrams for the PBE and PBE-D3 optimized structures (Fig. 6). The best agreement with WAXS diagrams, quantified using the same criteria as above, is found for (9,9) SiCH₃ and (11,11) GeCH₃ nanotubes (Fig. 6 and Supplementary Note 7). These results strengthen the proposed assignment, thereby supporting the adopted parameterization for the quadratic-energy-driven fitting procedure (Fig. 4) as well as the whole procedure developed for fitting WAXS diagrams.

In an attempt to elucidate the experimental findings, further geometry optimizations were carried out for several AC and ZZ SiCH₃ and GeCH₃ nanotubes. The DFT energy $E$ is calculated over a periodic unit cell of the nanotube, where ZZ $(N,0)$ and AC $(N,N)$ nanotubes both contain $2N$ imogolite structural units within a period. Direct comparison of the energy per imogolite unit ($E/2N$) between different nanotubes structures is thus meaningful. Regardless of the inclusion (PBE-D3) or neglect (PBE) of dispersion interactions, we find the $E/2N$ minima for SiCH₃ and GeCH₃ AC nanotubes to be substantially lower in energy than the corresponding minima for ZZ analogs (Fig. 7). That is, AC SiCH₃ and GeCH₃ nanotubes are computed to be energetically favored over their ZZ counterparts. The substantial differences between computed $E/2N$ minima for AC and ZZ SiCH₃ (PBE: −0.42 eV, PBE-D3: −0.37 eV) and GeCH₃ nanotubes (PBE: −0.29 eV, PBE-D3: −0.30 eV) explain the formation of AC nanotubes as determined experimentally (Fig. 3). The minimal deviations between PBE and PBE-D3 results on the energy favorability of the AC nanotubes indicate a negligible role for dispersion interactions in making the AC structure energetically favored. The lower energy of AC systems must therefore originate from a more favorable (less strained) chemical bonding for the (methylated) nanotubes in the AC geometry with respect to the ZZ structure.

Extensive geometrical analysis for the nanotube structures at the DFT-optimized periodicity (reported in the Supplementary Notes 8 and 9) reveals that AC and ZZ rolling modes have a different effect on the local bonding for GeCH₃ and SiCH₃ nanotubes. For the analysis we adopt the same radial atomic labeling as in refs. [34,52] that is H₁–C₂–Si₃(Ge₃)–O₄–Al₅–O₆–H₇ from the nanotube cavity to its outer surface (see Fig. 1). Energy favorability of the SiCH₃ AC nanotubes stems mostly from the O₄–Al₅ and Al₅–O₆ bonds, and O₄–Al₅–O₄ angles. Conversely, for GeCH₃ nanotubes, AC rolling becomes energetically favored

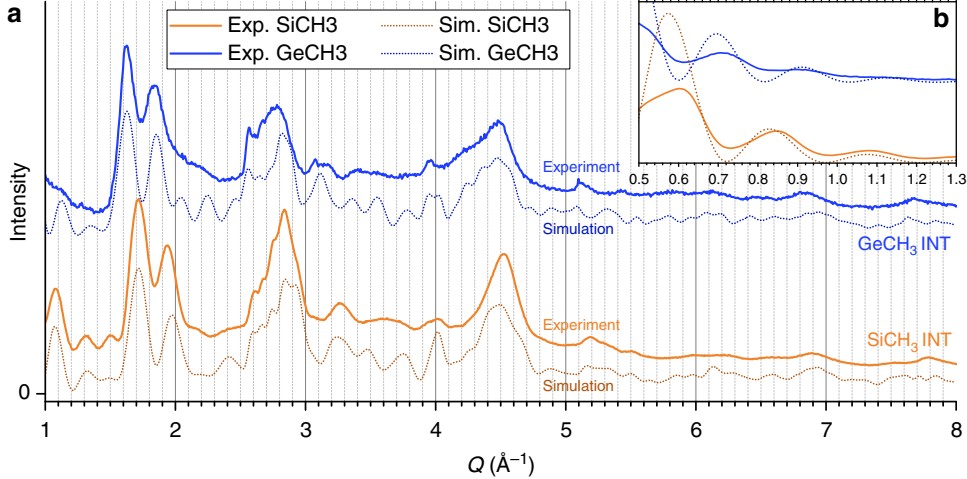

**Fig. 5** Comparison between experimental and calculated WAXS diagrams. **a** WAXS diagrams of methylated Si/GeCH₃ INT powders and related simulations. **b** The inset displays lower wave-vector range

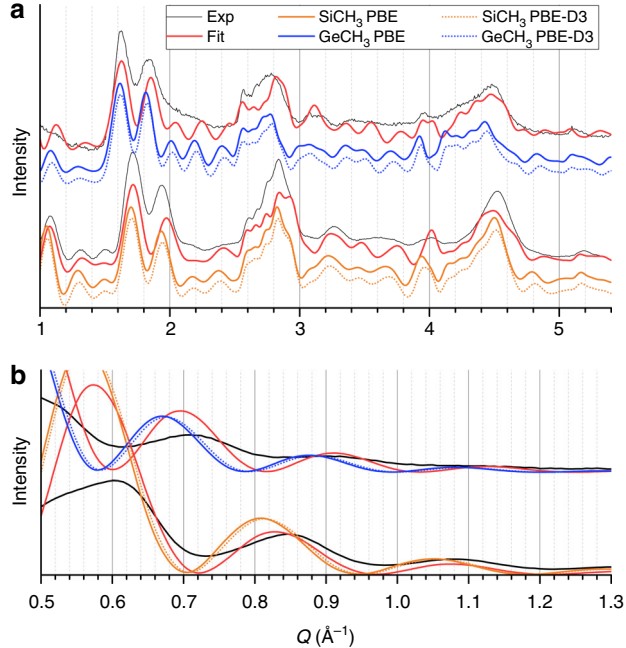

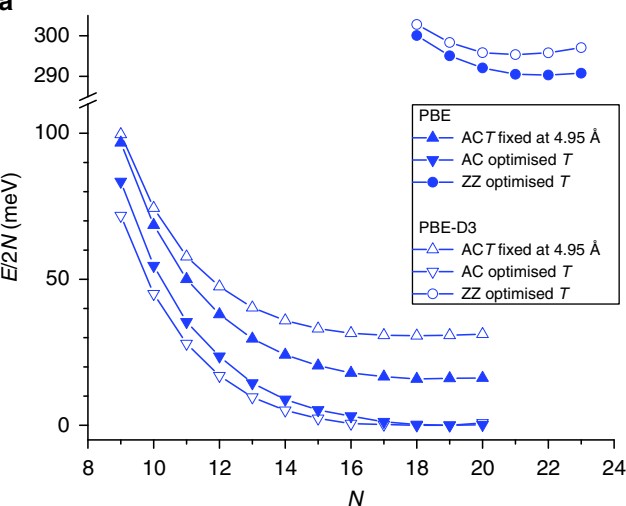

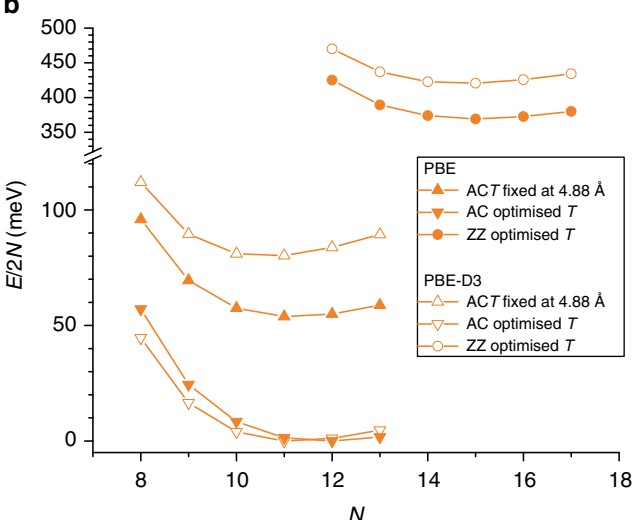

**Fig. 6** Calculated WAXS diagrams for DFT-relaxed imogolite nanotubes.
**a** Calculated WAXS diagrams for the DFT-optimized GeCH$_3$ and SiCH$_3$ INT structures at the experimentally fitted NT-periodicity, compared to experimental (Exp) and fitted (Fit) results. **b** Close-up on the small wave-vector region

**Fig. 7** Energy per imogolite unit for methylated nanotubes with different chiralities. Computed $E/2N$ profiles for AC and ZZ GeCH$_3$ (**a**) and SiCH$_3$ (**b**) nanotubes at PBE (filled symbol) and PBE-D3 (hollow symbol) level. AC traces have been computed for both the experimentally fitted and DFT-optimized NT-periods. ZZ traces are displayed only for DFT-optimized periods

mostly due to reduction of the distortion in the O$_6$–Al$_5$–O$_6$ angles, with negligible changes in the bond lengths. Direct confirmation of the less strained bonding of the gibbsite O$_4$–Al$_5$–O$_6$ layer, leading to an energetically more favorable bonding of the AC nanotubes with respect to the ZZ ones, is provided by analysis of the electronic structure of the nanotubes in Supplementary Note 10. The simulations reveal a substantially lower (as much as 0.3 eV) energy of the valence band for the AC nanotubes than for the ZZ ones. For the same number of imogolite units ($N$), the AC geometry results in a substantially more favorable environment for the electrons of the nanotube, contributing to lower the energy of the entire system.

Based on the PBE computed $E/2N$ profiles between hydroxylated ZZ SiOH and ZZ SiH nanotubes, the presence of an H-bonding network inside the nanotube cavity has been proposed to be crucial for the occurrence of a well-defined $E/2N$ minimum for INTs in Lee et al.[40]. However, substitution of the pendant silanol (−SiOH) group by (aprotic) phosphorous and arsenic derivatives is also computed (at TB-DFT level) to result in $E/2N$ profiles with well-defined minima[62]. In addition, the $E/2N$ profiles for ZZ and AC SiCH$_3$ in Fig. 7, and earlier DFT results for ZZ SiCH$_3$[52], also presents minima. Altogether these results indicate that, at least for Si-based INTs, the presence of an inner H-bonding network is not actually necessary for the appearance of a well-defined $E/2N$ minimum. The energy competition between strain of the Si$_3$-O$_4$ bond on the nanotube cavity, distortion of the gibbsite layer, and maximization of the H-bonding network on the outer layer (evidenced in Supplementary Figure 18) appear also to be effective in producing $E/2N$ profiles with minima, albeit less steep than for hydroxylated INTs[40]. The occurrence of progressively shallower $E/2N$ minima for ZZ and AC GeCH$_3$ nanotubes in Fig. 7 provides further evidence that the balance between the different sources of strain in m-INTs is clearly composition and radius dependent. Depending on the presence of Si or Ge, structural relaxation of m-INTs takes place

at different regions of the nanotube and with different energy gains. Supplementary Notes 10 and 11 report further electronic characterization of m-INTS. Inversion of the wall polarization between SiCH$_3$ and GeCH$_3$ INTs is highlighted. Moreover, potential interest of m-INTs for photocatalytic applications as well as electrostatic tuning of redox chemistry for confined molecules is discussed.

**The role of the synthesis environment.** In spite of the agreement between experimental and calculated WAXS diagrams for the DFT-optimized (9,9) SiCH$_3$ and (11,11) GeCH$_3$ nanotubes (Fig. 6), it is obvious in Fig. 7 that both PBE and PBE-D3 simulations fail in modeling these systems as $E/2N$ minima. In contrast to the WAXS fitted SiCH$_3$ (9,9) structure, PBE and PBE-D3 $E/2N$ minima are computed for the (12,12) and (11,11) nanotubes, respectively. The disagreement for GeCH$_3$ nanotubes is even larger with PBE and PBE-D3 computed $E/2N$ minima for the (19,19) structure, substantially far from the experimentally

fitted (11,11) value. The large deviations between experimentally fitted and energetically computed $N$, together with the aqueous synthesis environment for both SiCH$_3$ and GeCH$_3$[21,22], as well as the relatively high temperature of synthesis (90 °C), hint to a possibly over-simplistic nature of the computational models used, and to a non-negligible role for the nanotube interactions with the synthesis environment in fine-tuning the energy favored nanotube structure, an aspect previously documented experimentally for hydroxylated INTs[26].

It is interesting to note that, at PBE-D3 level, the $E/2N$ differences between experimentally fitted structures and computed minima are 0.015 and 0.03 eV for SiCH$_3$ and GeCH$_3$, respectively. Previous (force field) molecular dynamics simulations of water confined inside hydrophobic AC CNTs in the (5,5)–(20,20) range (diameter range: 0.7–2.7 nm, close to the values for the optimized INTs, see Supplementary Table 3–10) suggest changes up to about 0.1 eV (about 2.4 kcal mol$^{-1}$) in the free energy of nano-confined water molecules as a function of the CNT radius[63,64]. Assuming, given the hydrophobic nature of both CNTs and m-INTs, that quantitatively similar changes may be present for water nano-confined inside SiCH$_3$ and GeCH$_3$ nanotubes, confinement of no more than 1 residual water molecule from the aqueous synthesis environment every 3–7 imogolite units could be sufficient to turn the experimentally fitted (9,9) and (11,11) structures energetically favored over the computed PBE-D3 minima [(11,11) and (19,19) for SiCH$_3$ and GeCH$_3$, respectively]. While evidently speculative, we believe this argument deserves closer investigation. Further evidence of the role of the interactions with the synthesis environment in fine-tuning the m-INTs' structure and energy is provided by the non-negligible deviations between the experimentally fitted inner and outer radii (Supplementary Table 2) and the values computed for the AC nanotubes optimized in vacuo (Supplementary Tables 5, 6 and 9, 10). Given the current impracticability of DFT-based approaches to free energy sampling for m-INTs in synthesis aqueous environments owing to the size of the systems, we hope our results and considerations will stimulate interest in the subject by the force field and TB-DFT communities.

## Discussion

Whereas there has been remarkable experimental and theoretical progress in our knowledge about structure and properties of two-dimensional metal-oxide surfaces and interfaces[65], quantitative determination and understanding of the atomic structure of metal-oxide surfaces rolled into nanotubes lags behind. Here, the structure of two new members of the family of metal-oxide nanotubes, specifically single-walled methylated aluminosilicate and aluminogermanate nanotubes, has been determined at the atomic level from WAXS experiments, which is a first contribution to bridging the gap. We demonstrate that, unlike their $(N,0)$ ZZ hydroxylated analogs, methylated INTs roll up into a $(N,N)$ AC structure with $N = 9$ for SiCH$_3$ and $N = 11$ for GeCH$_3$ nanotubes. It follows that functionalization (methylation in the present case) of the INTs cavity not only offers control over the inner surface properties but it also leads to drastic structural changes such as change of the chiral vector of the nanotube. The results of the experimental WAXS fitting are supported by DFT simulations that predict AC rolling of methylated nanotubes to be energetically favored over ZZ structuring. $N$ values corresponding to minima of DFT energies for isolated nanotubes turn out to be larger than those deduced from WAXS fitting, indirectly confirming previous experimental suggestions of the role of the synthesis medium in fine-tuning the final diameter of INTs. Furthermore, our results rule out previously proposed models which are shown to be both inconsistent with WAXS results and

energetically disfavored, stressing the value of WAXS structure-determination for fundamental research in inorganic SWNTs.

More generally, this article introduces a singular fitting procedure to enable complete resolution of SWNTs atomic structures from WAXS diagrams. It is based, first, on the use of helical symmetries allowing one to consider the smallest unit cell and second, on semi-empirical energy minimization and ensuing reduction of the number of structural parameters to be fitted. The simple fitting approach proposed is directly applicable to the whole family of the imogolite-like metal-oxide nanotubes of geological and physico-chemical interest and whose structure is not solved precisely. It is also transferable to other large unit cell single-walled inorganic nanotubes currently synthesized and with potential applications in various fields. In the case of INTs, the unit cell is a slightly deformed hexagonal one but one should underline here that the use of helical symmetries imposes no constraint on the elementary cell[44]. Moreover, the role of the synthesis medium and the temperature, which is crucial in the synthesis of SWNTs in suspension[1,3], is included in the WAXS fitting procedure we propose, through the choice of the internal and external radii as fitting parameters.

## Methods

**Synthesis of methylated nanotubes.** The synthesis of Si or Ge nanotubes with a methylated inner cavity of nominal composition (OH)$_3$Al$_2$Si$_x$Ge$_{1-x}$CH$_3$ was performed according to the procedure described by Amara et al. (2015)[22]. Aluminum perchlorate (Sigma-Aldrich, 98%) was mixed in Teflon beakers with methyltriethoxysilane (MTES; Sigma-Aldrich, 99%) or methyltriethoxygermane (MTEG, ABCR, > 95%) for synthesizing either SiCH$_3$ INT ($x = 1$) or GeCH$_3$ INT ($x = 0$), respectively. The initial aluminum perchlorate concentration was set at $C = 0.1$ mol L$^{-1}$ and the concentration of MTEG/MTES was chosen so that the ratio [Al]/([Si] or [Ge]) is equal to 2. The obtained mixtures were slowly hydrolyzed by the addition of a 0.1 mol L$^{-1}$ NaOH (hydrolysis ratio [OH]/[Al] of 2), stirred overnight at room temperature, and then aged at 90 °C into an oven for 5 days. After recovering the suspensions, all samples were dialyzed against ultrapure water, using 8 kDa membranes in order to remove residual salts and alcohol in excess. Dialyses were performed until the conductivity dropped below 5 μS cm$^{-1}$.

**Sample preparation.** The dialyzed suspensions were dried at 60 °C during 24 h, the resulting sediment being milled in an agate mortar to obtain a fine powder. The obtained powders were held in cylindrical borosilicate glass capillaries (WJM-Glas, Müller GmbH, Germany) of 1 mm diameter that were flame-sealed.

**Wide-angle X-ray scattering.** The powder WAXS experiments have been carried out at the synchrotron SOLEIL (Gif-sur-Yvette, France) on the CRISTAL beamline. A monochromatic X-ray beam with a wavelength of $\lambda = 0.79176$ Å was extracted from the U20 undulator beam by means of an Si (111) double monochromator. Measurements were performed using a 21 perfect crystals Si (111) multi-analyzer allowing to access to a large range of wave-vector $Q$: 0.5Å$^{-1}$ < $Q$ < 8Å$^{-1}$ ($Q = \frac{4\pi}{\lambda}\sin(\theta)$). The high resolution (10$^{-3}$ Å$^{-1}$) provided by Si crystals is negligible in comparison to the modulation of imogolite WAXS diagrams even for the determination of the value of the period $T$ along the nanotube axis.

**DFT simulations.** All the DFT simulations were performed with the CP2K/Quickstep package[66], using the PBE[67] approximation to the exchange and correlation functional. Where used, van der Waals corrections were applied according to Grimme's DFT-D3 approach[68]. Core electrons were described with norm-conserving Goedecker, Teter, and Hutter pseudopotentials[69]. Valence electron Kohn-Sham states were expanded in terms of Gaussian functions with molecularly optimized double-ζ polarized basis sets (m-DZVP), which ensures a small basis set superposition error[70]. For the auxiliary basis set of plane waves a 320 Ry cutoff was used. Reciprocal space sampling was restricted to the Γ point. The adopted convergence thresholds for the geometry optimizations were 10$^{-4}$ Ha Bohr$^{-1}$ on the maximum atomic force, and 3 × 10$^{-4}$ Ha Bohr$^{-1}$ on the root mean square residual of all the atomic forces. Calculations were performed using periodic boundary conditions (>20 Å vacuum-buffer perpendicularly to the nanotube axis) with both experimentally derived and DFT-optimized values of the nanotube period along its axis. Optimized period values at PBE (PBE-D3) level are 4.87 Å (4.83 Å) for AC GeCH$_3$ INTs, 4.72 Å (4.72 Å) for AC SiCH$_3$ INTs, 8.50 Å (8.45 Å) for ZZ GeCH$_3$ INTs, and 8.54 Å (8.50 Å) for ZZ GeCH$_3$ INTs.

**Data availability.** The main result of this study is the WAXS diagrams' fitting process whose details are given in Supplementary Note 3. Thus, the corresponding

code, which was written without a friendly-user interface, is not publicly available but it is available from the corresponding authors on reasonable request.

The atomic structure of SiCH₃ and GeCH₃ INTs are deposited at the Cambridge Crystallographic Data Centre under CCDC accession codes: 1838953, 1838955. These data can be obtained free of charge from The Cambridge Crystallographic Data Centre via www.ccdc.cam.ac.uk/data_request/cif [www.ccdc.cam.ac.uk/data_request/cif].

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

## Acknowledgements

J.D.E., E. Poli, and G.T. acknowledge support from EPSRC UK (EP/I004483/1, EP/K013610/1, and EP/P022189/1) and G.T. also acknowledges Université Paris Sud for award of a Visiting Professorship at the Laboratoire de Physique des Solides in 2016. M.S.A. acknowledges for award of Temporary Lecturer and Research assistant funding of Université Paris Sud. This work made use of the Beijing Computational Science Research Center, ARCHER (via the UKCP Consortium, EP/K013610/1), and the STFC Hartree Centre (Daresbury Laboratory, UK) High Performance Computing facilities. G.M. and P.L. acknowledge Stéphane Rols (Institut Laue Langevin, France) and Benjamin Rotenberg (Phenix laboratory, France) for interesting exchanges on empirical potential models. The authors acknowledge Erik Elkaim for his local contact role on beamline CRISTAL (synchrotron SOLEIL) as well as for interesting discussions.

## Author contributions

The study was designed by P.L., together with G.T. for the DFT part. Imogolite nanotubes were synthesized by M.S.A. and E. Paineau. M.S.A., G.M., E. Paineau, P.L., and S.R. performed WAXS experiments. G.M., MSA, E.Paineau, S.R., and P.L. analyzed the experimental data. Structural minimization was carried out by G.M. and P.L. for the geometrical part and by Z.C. and J.D.E. for ab-initio one. Z.C., J.D.E., and E. Poli prepared the inputs and executed DFT simulations. Analysis of DFT results was carried out by Z.C., J.D.E., E. Poli, L.-M.L., and G.T. G.M., P.L., and G.T. wrote the first draft of the manuscript. All authors have contributed to the elaboration of the manuscript and have given approval to the final version.

## Additional information

**Competing interests:** The authors declare no competing interests.

