## [Peer Review File · Nature Communications]

REVIEWERS' COMMENTS:

Reviewer #1 (Remarks to the Author):

The paper "Structural resolution of inorganic nanotubes with complex stoichiometry" by Launois et al is an impressive paper concerning the methodology to solve atomic structure of nanotubes from powder scattering patterns. The exploitation of this data enlightens the electrostatic forces as well as the electronic structure. The paper can be published after some shortening of the exploitation paragraphs.

The originality of the paper is the structural refinement of the powder X-ray pattern that includes a molecular mechanic step. This powerful in the case of the nanotubes. There is only one example in the case of organic crystal (Oda et al, JACS, 2008).

The DFT optimization part of the paper should be more summarized.

Sentence such as " for the specialist reader" or "to the best of our knowledge" are not necessary.

In conclusion, the paper is excellent because of methodology for solving the structure as well as the deep analysis that gives clear information on electronic properties of these inorganic nanotubes.

Reviewer #2 (Remarks to the Author):

This paper presents the structure determination using Wide angle X-ray scattering of methylated imogolite nanotubes. Both the experimental and computational methodology appear rigorous and all the details to ensure reproducibility of this research are reported. A new fitting procedure that allows resolving SWNT structures from WAXS diagrams is proposed. Overall, the quality of the paper is high.

The method: it is proposed that the presented approach could be applied to structure determination of other inorganic nanotubes. This is indeed a hot topic as inorganic nanotubes, especially mineral ones, can have complex stoichiometry and achieving full knowledge of their atomic structure is in many cases challenging. Moreover, inorganic and mineral nanotubes are of high interest in the fields of gas storage and catalysis and a method that allows extracting their detailed structure would most certainly be highly beneficial to these fields.

The results for methylated imogolite: While I do not think this structure determination itself is of enough interest for publication on Nature Communications, It is a very interesting example to show the proposed approach for structure determination. What keeps imogolite at a specific diameter and chirality seems to be the inner wall hydrogen bonding, a network that gets destroyed when the inner OH are substituted with CH₃. Therefore it is not too surprising that the tubes can assume other chiralities. In particular, the armchair configuration brings the OH units far from each other, so again I am not surprised that substituting OH with CH₃ would favour an armchair configuration as the repulsion between the fragments would be minimised.

What really makes this paper of high, wide and general interest is the new approach proposed for inorganic nanotube structure determination. However, I feel I have not enough expertise in the use of WAXS to judge whether the novelty of this application is such to be published on Nature Communications.

I have a few other minor comments

I would not refer to imogolite as metal oxide, but rather as aluminosilicate and aluminogermanate.

Imogolite are not the first INTs to be discovered. There are many studies on other inorganic nanotubes before 1962, eg.

Pauling (1930) The structure of chlorites, PNAS

Bates et al (1950) Tubular Crystals of Chrysotile Asbestos, Science. 111, 512

Bates et al. (1950) Morphology and structure of endellite and halloysite, Am. Miner. 35, 463

Waser (1955) Fourier transforms and scattering intensities of tubular objects, Acta Cryst. 8, 142

A number of studies have provided atomic positions for the non substituted imogolite, both as silicate and germanate.

Dear Editor,

We have made revisions based on the suggestions of the reviewers, as it is explained below (text in black, reviewers' comments being in blue).

We would like to thank the reviewers for their interest in our manuscript, as well as for their suggestions for improving it. We took into account all reviewers' suggestions.

We propose here a minor modification of the two-sentence editor's summary: 'Structural determination of inorganic nanotubes has lagged far behind that of their carbon-based counterparts. Here, the authors present a transferable methodology, combining wide angle X-ray scattering and computation, to quantitatively resolve the atomic structure of inorganic nanotubes with complex stoichiometry.'

Please also note that we wish to opt in to the publication of the reviewers' reports.

Yours sincerely,

Pascale Launois and Gilberto Teobaldi

Reviewer #1 (Remarks to the Author):

We thank the reviewer for his (her) positive assessment, for his (her) proposals for improving the readability of the article.

The paper "Structural resolution of inorganic nanotubes with complex stoichiometry" by Launois et al is an impressive paper concerning the methodology to solve atomic structure of nanotubes from powder scattering patterns. The exploitation of this data enlightens the electrostatic forces as well as the electronic structure. The paper can be published after some shortening of the exploitation paragraphs.

The originality of the paper is the structural refinement of the powder X-ray pattern that includes a molecular mechanic step. This powerful in the case of the nanotubes. There is only one example in the case of organic crystal (Oda et al, JACS, 2008).

A sentence was added in the manuscript to refer to this article: 'One cannot take advantage of three-dimensional crystalline order, as was done recently by Oda and co-workers to solve the molecular structure of self-assembled organic nanoribbons [Reiko Oda, Franck Artzner, Michel Laguerre and Ivan Huc, Molecular Structure of Self-Assembled Chiral Nanoribbons and Nanotubules Revealed in the Hydrated State, *J. Am. Chem. Soc.*, 2008, 130 (44), pp 14705–14712, DOI: 10.1021/ja8048964]'

The DFT optimization part of the paper should be more summarized.

Sentence such as "for the specialist reader" or "to the best of our knowledge" are not necessary.

As requested we have removed instances of "for the specialist reader" and "to the best of our knowledge". Following the Reviewer's comment, we have streamlined the DFT-optimization part of the paper moving more technical aspects and electronic structure characterization to the Supporting Information, thus reducing the DFT section by 27 lines (i.e. over one manuscript page). Specifically,

Page 14: "For the specialist reader we note that, to the best of our knowledge, no comparison between E/2N profiles (or analogous strain vs. radius analysis) has ever been computed or presented for methylated AC and ZZ nanotubes in the existing literature." has been removed.

Page 16: “For the specialist reader we note that, to the best of our knowledge, comparison between the electronic structure of AC and ZZ m-INTs has not been previously reported in the literature.” has been removed.

Pages 16-17: “The computed bond distances for the inner pendant groups [H_1-C_2 , $C_2-Si_3(Ge_3)$, and $Si_3(Ge_3)-O_4$ bonds] are minimally affected by AC or ZZ rolling, with deviations smaller than 5×10^{-3} Å (Supplementary Figures 12 to 14). The latter results clearly indicate that relaxation of the inner methyl groups and adjacent $C_2-Si_3(Ge_3)-O_4$ tetrahedron does not play any major role in lowering the energy of AC nanotubes with respect to ZZ analogs.

The over 0.1 Å shorter hydrogen bonding distances on the outer surface of the AC $SiCH_3$ nanotubes with respect to the ZZ ones (Supplementary Figure 18) provides further energetic favorability to AC rolling over the ZZ geometry. The substantially smaller (<0.02 Å) differences in hydrogen bonding distances on the outer surface of AC and ZZ $GeCH_3$ nanotubes indicate this additional mechanism of stabilization is not present for larger-diameter (reduced curvature) nanotubes, reiterating that depending on the presence of either Si or Ge the geometric origins of AC stabilization are different.” has been moved to the Supporting Information (Supplementary Note 8).

Page 17: “On a specialist level, we note that, based on the PBE computed E/2N profiles...” has been amended to read:

“Based on the PBE computed E/2N profiles...”

Page 19: The “The electronic structure of armchair m-INTs” subsection has been moved to the Supporting Information (Supplementary Note 10 and 11). Main results are presented as follows: “Supplementary Notes 10 and 11 report further electronic characterization of m-INTs. Inversion of the wall polarization between $SiCH_3$ and $GeCH_3$ INTs is highlighted. Moreover, potential interest of m-INTs for photocatalytic applications as well as electrostatic tuning of redox chemistry for confined molecules is discussed”.

Page numbering refers to the first version of the manuscript.

In conclusion, the paper is excellent because of methodology for solving the structure as well as the deep analysis that gives clear information on electronic properties of these inorganic nanotubes.

Reviewer #2 (Remarks to the Author):

We thank the reviewer for his (her) comments.

This paper presents the structure determination using Wide angle X-ray scattering of methylated imogolite nanotubes. Both the experimental and computational methodology appear rigorous and all the details to ensure reproducibility of this research are reported. A new fitting procedure that allows resolving SWNT structures from WAXS diagrams is proposed. Overall, the quality of the paper is high.

The method: it is proposed that the presented approach could be applied to structure determination of other inorganic nanotubes. This is indeed a hot topic as inorganic nanotubes, especially mineral

ones, can have complex stoichiometry and achieving full knowledge of their atomic structure is in many cases challenging. Moreover, inorganic and mineral nanotubes are of high interest in the fields of gas storage and catalysis and a method that allows extracting their detailed structure would most certainly be highly beneficial to these fields.

The results for methylated imogolite: While I do not think this structure determination itself is of enough interest for publication on Nature Communications, it is a very interesting example to show the proposed approach for structure determination. What keeps imogolite at a specific diameter and chirality seems to be the inner wall hydrogen bonding, a network that gets destroyed when the inner OH are substituted with CH₃. Therefore it is not too surprising that the tubes can assume other chiralities. In particular, the armchair configuration brings the OH units far from each other, so again I am not surprised that substituting OH with CH₃ would favour an armchair configuration as the repulsion between the fragments would be minimised.

What really makes this paper of high, wide and general interest is the new approach proposed for inorganic nanotube structure determination. However, I feel I have not enough expertise in the use of WAXS to judge whether the novelty of this application is such to be published on Nature Communications.

I have a few other minor comments

I would not refer to imogolite as metal oxide, but rather as aluminosilicate and aluminogermanate.

As requested, imogolite is now referred to as aluminosilicate and aluminogermanate (instead of metal oxide) through the manuscript.

Imogolite are not the first INTs to be discovered. There are many studies on other inorganic nanotubes before 1962, eg.

Pauling (1930) The structure of chlorites, PNAS

Bates et al (1950) Tubular Crystals of Chrysotile Asbestos, Science. 111, 512

Bates et al. (1950) Morphology and structure of endellite and halloysite, Am. Miner. 35, 463

Waser (1955) Fourier transforms and scattering intensities of tubular objects, Acta Cryst. 8, 142

The reviewer reminds us, rightly, that other inorganic nanotubular forms, such as halloysite or chrysotile asbestos, were known before imogolite nanotubes. We have not discussed them in our article because it is focused on the methodology to solve the structure of *single-walled* nanotubes with complex stoichiometry. It is a challenging issue because X-ray scattering diagrams to analyze are made of broad peaks (in contrast to X-ray scattering diagrams of multi-walled nanotubular structures with lateral crystalline order over a sufficiently large width). But we have added, in the paragraph 'The structure of imogolite nanotubes: state of the art', a reference to a recent review article presenting an historical perspective similar to the one suggested by reviewer 2 [*Clay mineral nanotubes: stability, structure and properties*, H.A. Duarte et al, Intechopen, 2012; DOI: 10.5772/34459]. The total number of references being limited to seventy, the reference about smoothing and differentiation of data by Savitsky and Golay has been removed.

A number of studies have provided atomic positions for the non substituted imogolite, both as silicate and germanate.

Such studies are cited in our article but maybe not in a sufficient explicit way. We have thus made it clearer in page 4, where we now write "Quantitative interpretation of imogolite WAXS diagrams has not been achieved yet, despite intensive research *and while atomic positions for (OH)₃Al₂O₃Si(Ge)OH INTs are provided in numerical simulations articles [Demichelis 2010, Lee 2011].*" (part of the sentence in italics has been added).